

# Deep-learning based climate downscaling using the super-resolution method: a case study over the western US

Xingying Huang[1]*

1Earth Research Institute, University of California Santa Barbara, Santa Barbara, 93106, USA
*Correspondence to: Xingying Huang (xingyhuang@gmail.com)

**Abstract.** Demand for high-resolution climate information is growing rapidly to fulfill the needs of both scientists and stakeholders. However, deriving high-quality fine-resolution information is still challenging due to either the complexity of a dynamical climate model or the uncertainty of an empirical statistical model. In this work, a new downscaling framework is developed using the deep-learning based super-resolution method to generate very high-resolution output from coarse-resolution input. The modeling framework has been trained, tested, and validated for generating high-resolution (here, 4 km) climate data focusing on temperature and precipitation at daily scale from the year 1981 to 2010. This newly designed downscaling framework is composed of multiple convolutional layers involving batch normalization, rectification-linear unit, and skip connection strategies, with different loss functions explored. The overall logic for this modeling framework is to learn optimal parameters from the training data for later-on prediction applications. This new method and framework are found to largely reduce the time and computation cost (~23 milliseconds for one-day inference) for climate downscaling compared to current downscaling strategies. The strength and limitation of this deep-learning based downscaling have been investigated and evaluated using both fine-scale gridded observations and dynamical downscaling data from regional climate models. The performance of this deep-learning framework is found to be competitive in either generating the spatial details or maintaining the temporal evolutions at a very fine grid-scale. It is promising that this deep-learning based downscaling method can be a powerful and effective way to retrieve fine-scale climate information from other coarse-resolution climate data. When seeking an efficient and affordable way for intensive climate downscaling, an optimized convolution neural network framework like the one explored here could be an alternative option and applied to a broad relevant application.



## 1 Introduction

With the increasing demand for high-resolution climate data across emerging climate studies and real-world needs (Giorgi et al., 2009; Stocker et al., 2014; Roberts et al., 2018), rapidly growing efforts have focused on developing methods and techniques to retrieve fine-scale details from coarse-resolution source either from reanalysis or simulations (Wood et al., 2004; Maraun et al., 2010; Giorgi and Gutowski, 2015). Existing downscaling methods mainly include but not limited to traditional dynamical downscaling (using either regional climate models, variable-resolution global climate modeling, or high-resolution global climate models), and empirical statistical downscaling (either linear or nonlinear), attributing with unique strengths and also limitations (Huang et al., 2016). In detail, traditional dynamical downscaling relies on a complex numerical model, with relatively costly computation and time efforts, needing physical schemes optimizations. Statistical downscaling is relatively resource-efficient, but generally be constrained to the assumptions of temporal stationarity, empirical knowledge of the controlling factors/predictors, and/or perfect prognostic bias correction.

In recent years, machine learning has gained its popularity in climate science (Liu et al. 2016; Kurth et al. 2018; Rasp et al. 2018; Rolnick et al., 2019; Tran Anh et al. 2019; Ahmed et al. 2020). The area of climate downscaling also sees some preliminary applications (Vandal et al. 2017; Rodrigues et al. 2018; Chang et al., 2018; Pan et al, 2019). For example, Vandal et al. (2017) presented a DeepSD framework composed of three convolutional layers. And Rodrigues et al. (2018) explored climate downscaling using several convolution layers and locality-specific layers. Further, Pan et al (2019) used convolutional layers and fully connected layers, with every entry in the input being connected to every entry in the output regardless of their locations, to predict per-grid point value. Overall, former studies exhibited the possibility of using deep-learning for climate downscaling compared to traditional statistical downscaling, but still leaving large space for in-depth explorations. Previous studies mostly used basic and early-stage deep learning strategies with simple convolutional neural frameworks and showed only a few experiments with downscaling results at moderate grid resolutions over very limited study regions.

Importantly, deep learning has advanced a lot since then and is becoming rather sophisticated for many more applications (Lecun et al. 2015; He et al. 2016; Oord et al. 2016; Silver et al. 2017; Devlin et al. 2018;). In this work, the main goal is to explore the construction and application of a comprehensive deep-learning based framework for retrieving fine-scale temperature and precipitation data, using the image super-resolution method. In the area of image processing, image super-resolution is used to reconstruct high-resolution images from low-resolution images, which has advanced significantly in recent years using deep learning methods (Ledig et al. 2017; Lim et al. 2017; Zhang et al. 2018; Yang et al. 2019; Wang et al. 2020; Zhang et al. 2020). Climate downscaling has similar goals in terms of generating high-resolution information and has been an important topic for decades. The present study aims to incorporate up-to-date deep learning schemes for network design with robust tests of layers composition, layer connections, and loss functions. This work uses cutting-edge training strategies with high-performed GPUs for large numbers of epochs (as detailed in the methods). The modeling framework has been trained, tested, and validated for generating high-resolution (here, 4 km) near-surface temperature and precipitation data from coarse input (~81km) at



daily scale from the year 1981 to 2010. Comprehensive analysis of the results is presented compared to not only the
"ground-truth" observations but also available traditional dynamical downscaling data. Overall, this study shows the
promise of a broad application using the deep-learning based modeling framework for climate downscaling.

**2 Methods and dataset**

**2.1 Building deep-learning neural network for downscaling**

The logic of using a deep convolution neural network framework for downscaling is to take coarse-scale input and/or
supporting data to produce fine-scale output/prediction (as depicted in Figure 1). To build this framework, during the
training stage, the finalized construction has incorporated convolutional layers (Conv), rectification linear unit (ReLU)
layers), and batch normalization layers (BN) as detailed below.

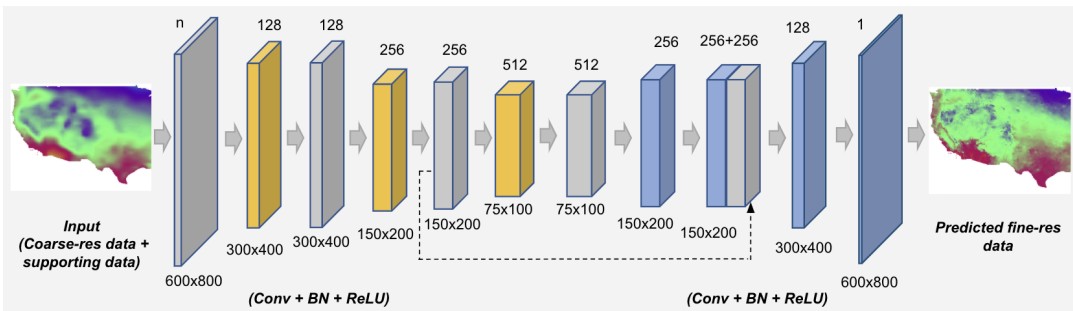


**Figure 1: Overview of the deep convolutional neural network and its components used in this study.** The network
is composed of a sequence of convolutional layers, batch normalization layers, and ReLU layers. (Note: Dashed arrow
lines represent the skip connections; Gold-colored layers refer to the ones with stride value of 2. The numbers on the
top of each convolutional layer refer to the filter size, and the number on the bottom of that refers to the grid size of
the image.)

**2.1.1 Convolutional layers**
In short, a convolution layer consists of a certain number of filters. Each filter (i.e. channel) will operate on a local
region (e.g. 3x3 region) within the input dimension in a sliding window manner. This operation is called convolution.
Each convolution has two groups of trainable parameters: i.e. weights ($W$) and bias ($b$), and these trainable parameters
will be learned in the training stage. Here, the author used 128, 256, and 512 filters (corresponding to the number of
output channels) for different layers. Each filter has parameters: weight $W$ and bias $b$. Supposing the input is with $M$
channels and denoted as $X_{(M)}$, and the filter number is $N$, for a $K \times K$ sliding window (here, $K = 3$) centered at $(i, j)$,
the convolution output for the $n$th filter is computed in the following way:



$$Y_{i,j}^n = \sum_{0}^{M-1} \sum_{=0}^{K-1} \sum_{b=0}^{K-1} (W_{a,b}^{n,m} X_{(i-K/2+a,\, i-K/2+b)} + b^n)$$

here, $Y_{i,j}^n$ represents the $n$th filter's output at a location $(i,j)$. When designing a convolution layer, the parameters for stride and padding also need to be specified. The stride value controls the offset of the sliding window when moving to the next sliding. Padding is used to pad extra values (usually set as 0) at the borders to gather enough data for the convolution operation on the entries centered at borders.

In this framework (Figure 1), stride value of 1 and stride value of 2 are used for different convolutional layers. For the convolutional layer with stride being 1, the spatial domain size of the output is the same with the spatial size of the input. For the convolutional layer with stride set as 2, the spatial size of the output is smaller (around half in each dimension) than that of the input. In deep learning practices, convolutional layers with stride being 2 are used to increase the receptive fields of the convolutional layers. The receptive field is defined as the area where the convolutional filters can influence. Usually, the area where a single convolution filter can influence depends on its kernel size (here, 3x3). The area (or receptive field) is accumulated by using more convolutional layers and having stride be 2 will further accumulate the receptive field.

The left part of the network, which transforms the input into smaller dimension features, is called encoder as a custom in the computer vision field. In the encoder part, six convolutional layers are used with three of them having stride values of 2 and the rest three having stride values of 1. Each convolutional layer is connected by a batch normalization layer and a ReLU layer as explained below. Accordingly, the right part of the network, which transforms the lower dimensional features to final output, is called decoder. In the decoder part, there are three convolutional layers and three nearest upsampling layers with the hard-coded nearest upsampling rules (not shown in Figure 1, as no parameter needs to be trained for them). The upsampling layer and convolutional layer are applied alternately. Each convolutional layer in the decoder part is also connected by a batch normalization layer and a ReLU layer. In total, 28 layers have been utilized.

**2.1.2 ReLU layers**

As the mathematical operation of a convolution layer is a linear function, nonlinear functions are needed between the convolution layers to let the entire network describe a non-linear mapping. These nonlinear functions are called activation functions. In deep learning field, ReLU is a widely used activation functions as a one-to-one mathematical operation, defined as: $ReLU(x) = max(0,x)$. This simple approximation can not only compute fast but also match the capability of more complex nonlinear functions, received as a foundation for current deep learning models (Nair et al. 2010, Glorot et al. 2011, Krizhevsky et al. 2012). In general, the key to a successful deep-learning based framework is to approximate the highly complex relation (here, for fine-scale temperature and precipitation features) by combining a sequence of linear and non-linear operations.

**2.1.3 Batch normalization layers**





Deep neural networks can be sensitive to the initialized values of the trainable parameters during the training process.
To reduce such sensitivity, batch normalization layers are used as a common way to stabilize the training, which is
used to re-center and rescale the input and has been shown to be effective in improving the training speed, accuracy,
and stability of deep neural networks (Ioffe et al. 2015). The batch normalization is computed through three steps: a)
Calculate the mean ($\mu$) and standard deviation ($\sigma$) of the input; b) Subtract the mean from the input and divide it with
the standard deviation (i.e. $\hat{x} = (x - \mu)/\sqrt{\sigma^2_{+\epsilon}}$); c) Using the results from step b) to multiply the batch normalization
layer's parameter $\gamma$ and then added to the layer's parameter $\beta$(i.e. $y = \gamma\hat{x} + \beta$). Detailed equations and algorithms can
be found in Ioffe et al. (2015).

### 2.1.4 Skip connections

The idea of skip connections is to concatenate the outputs from two non-consecutive layers. Previous work shows that
skip connections can improve some details for the output (Ronneberger et al. 2015). In this framework, two skip
connections (see the stacked layers in Figure 1) are used as seen fit.

### 2.2 Network training: loss function selection


In the training stage, the difference between the prediction and the target is used to guide the updates of the trainable
parameters. The mathematical function to compute such a difference is called loss function. Several commonly used
loss functions have been tested to train a network based on existing successful applications in computer vision (Zhao
et al. 2016; Johnson et al. 2016; Liu et al. 2018). In this work, two types of widely-used loss functions are chosen: L2
loss and L1 loss.

L2 or L1 loss is defined as the mean square error or the absolute difference loss between the prediction and the target
respectively, i.e.:

$$L_2 = \frac{(I^{pred} - I^{gt})^2}{N_{I^{gt}}} \; ; \; L_1 = \frac{|I^{pred} - I^{gt}|}{N_{I^{gt}}}$$



The derivative of L2 and L1 loss are:

$$\frac{dL2}{dI^{pred}} = \frac{2(I^{pred} - I^{gt})}{N_{I^{gt}}} \; ; \; \frac{dL1}{dI^{pred}} = 1 \text{ (if } I^{pred} - I^{gt} > 0), \; \frac{dL1}{dI^{pred}} = -1 \text{ (if } I^{pred} - I^{gt} < 0)$$



Usually, L2 loss is more sensitive to the scale of difference between prediction and ground truth than L1 loss, while
L2 loss's derivative is continuous at the value 0 while L1 loss's derivative is not. As the derivative of the loss function
is used to determine the updated values for the model parameters during the training process, the update of the model
parameters is not stable around the value 0 for L1 loss.





In the case of precipitation downscaling, most of the entries are zeros. As a result, using L1 loss is more difficult to converge to an optimal solution than using L2 loss. Therefore, the output trained with L2 loss is used as the prediction for precipitation in this study. While, in the case of temperature downscaling, the input values are continuous without zero values (in the unit of Kelvin). The model using L1 loss has less chance for suffering instability, and L2 loss's derivative is sensitive to the scale of the difference between prediction and ground truth. Therefore, for the temperature downscaling results, the model using L1 loss has a larger chance to converge more efficiently and reach a final optimal solution than L2 loss. As a result, the output trained with L1 loss is used as the prediction for temperature.

## 2.3 Dataset and computation

### 2.3.1 Dataset

Daily data is targeted covering the whole western US from 1981 to 2010. The goal is to downscale the coarse-resolution reanalysis input (here, using ERA-interim, ~81 km) to 4 km (also the resolution of the ground-truth) for near-surface (2 m) temperature (T2) and precipitation (Pr) for each day. A summary of the dataset used in this study is given in Table 1. In detail, ERA-interim, a widely-used reanalysis dataset (Dee et al., 2011), is chosen as the coarse-resolution input. A well-received high-quality gridded observational dataset, PRISM (Parameter-elevation Regressions on Independent Slopes Model, Daly et al., 2008), is applied as the ground truth for training purposes.

**Table 1: Dataset description as used in this study**

| Data type | Dataset source | Spatial resolution | Time periods and frequency | Variables |
|---|---|---|---|---|
| **Coarse input** | ERA-interim | ~81 km | 1981-2010; Daily | T2, Pr |
| **Ground-truth (i.e. target)** | PRISM | 4 km | 1981-2010; Daily | T2, Pr |
| **Supporting data** | ERA-interim | ~81 km | invariant | Elevation |
| **Supporting data** | PRISM | 4 km | invariant | Elevation |
| **Supporting data** | ERA-interim | ~81 km | 1981-2010; Daily | U, V, RH, Q (all at 850 hPa) |





| | | | | |
|---|---|---|---|---|
| **Output** | Training | 4 km | 1981-1990; Daily | T2, Pr |
| **Output** | Prediction | 4 km | 1991-2010; Daily | T2, Pr |
| **Dynamical downscaling dataset** | NA-CORDEX (WRF forced by ERA-interim) (Mearns et al., 2017) | 25 km | 1991-2010; Daily | T2, Pr |
| **Dynamical downscaling dataset** | WRF high-resolution downscaling (forced by NARR) (Liu et al., 2017) | 4 km | 2001-2010; Daily | T2, Pr |

Supporting datasets have been used together with the input for network training purposes, including elevations from both input and ground-truth sources at different native grid resolutions for both temperature and precipitation downscaling. Due to the discontinuity and complexity of precipitation field, additional supporting datasets have been used in addition to the elevations, including zonal and meridional winds (U and V), relative humidity (RH), and specific humidity (Q) from the coarse-resolution input data (i.e. ERA-interim) at 850 hPa vertical level. For evaluation purpose, dynamical downscaling datasets are also used, including public-shared WRF (the Weather Research and Forecasting model, Skamarock et al., 2008) simulations at 25 km from NA-CORDEX (Coordinated Regional Climate Downscaling Experiment featuring simulations for North America) (Mearns et al., 2017) and WRF simulations at 4 km covering the CONUS (Contiguous United States) (Liu et al., 2017), as described in Table 1. For analysis' convenience, all the datasets have been regridded to 4 km using the bilinear method.

**2.3.2 Training and inference**

To train the network, daily input in the first 10 years (i.e. 1981-1990) is used as training data and the rest 20 years are used for testing (i.e. for prediction and validation). The study region covers the whole western US regions, with a domain of 600x800 grid boxes after being re-gridded to 4 km. The network has been trained with 800 epochs, and each epoch refers to the full training of all the 10 years' data. During the training iteration process, the adjustment of model parameters is controlled by the learning rate. Here, the learning rate is set as 0.002 for the first 600 epochs and 0.0002 for the remaining 200 epochs. The last epoch from the trained model is used for the prediction. The determined



model framework with optimized parameters is then used to perform the inference on the remaining years' values (i.e. 1991-2010).

The PyTorch framework is used to build deep learning models. To speed up the dataset reading, the training data has been converted to HDF5 database format, which provides a faster query compared to the NetCDF format or other non-database files. The total trainable parameter number is ~7,500,000. The training loss curve from the finalized downscaling framework is shown in the supplemental (Figure S1).

**2.3.3 Computation and time cost**

This study has used 8 NVIDIA GPUs together to train the models for 800 epochs. Each training time is around 22 hours on 10 years' dataset. The inference (i.e. testing) time for one-day data is 22.75 milliseconds, and each day has been predicted in parallel using the trained model. The training time and inference time could be longer or shorter depending on what types of GPUs to use.

**3 Results**

**3.1 Temperature**

Firstly, the prediction performance for the yearly average temperature is shown (Figure 2). It can be seen that the prediction results (hereafter, named as Supres) closely match the ground-truth (i.e., the PRISM observations), in terms of both spatial patterns and the grid-scale values. The spatial correlation is about ~0.997 between Supres and PRISM, while the coarse input and WRF 4km also show a high correlation with PRISM for about ~0.98 to 0.99. Given near-surface temperature is strongly elevation-dependent, the supporting dataset of elevations from input and target (i.e. the ground-truth) (Figure S2), provides significant information for neural network learning to reconstruct the spatial details for the temperature over complex terrains due to associated orographic effects.



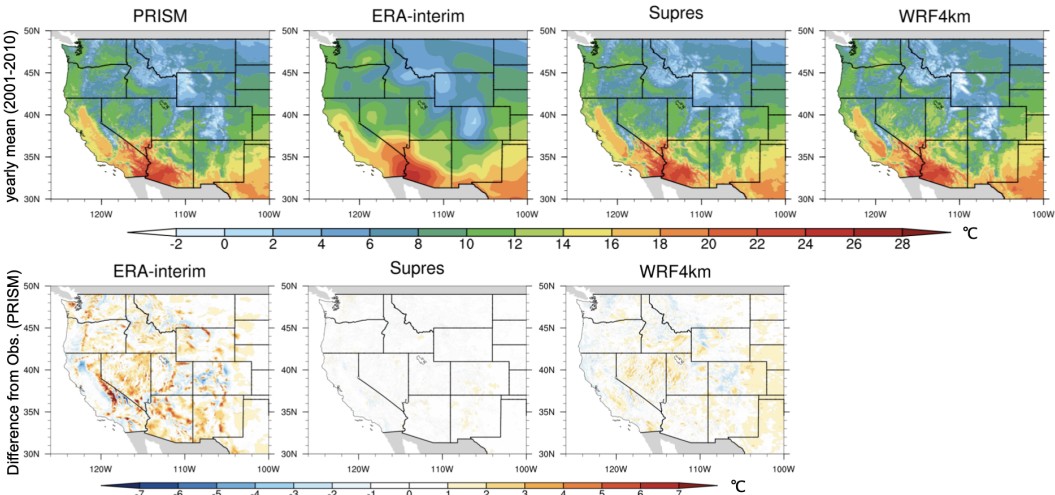

244

**Figure 2: Yearly average near-surface (2m) temperature (T2) over 2001-2010. Upper row:** T2 mean from PRISM
("ground-truth"), ERA-interim (input), Supres (i.e. the deep-learning based prediction), and WRF 4km; **Bottom row:**
Absolute differences from PRISM for the input, Supres, and WRF 4km results.

When zoomed into California, where diverse climate divisions locate (including coastal, inland valley, complex
mountainous, desert, etc.), the fine-scale spatial features in the prediction show notable improvement from coarse
resolution input. When compared to the WRF 4km dynamical downscaling data, the prediction outperforms if
comparing the differences from the ground-truth (Figure 2). The mean absolute differences from the
observation/reference are around 0.93K, 0.34K, and 0.71K for input, Supres, and the WRF 4km, respectively. A
supplemental comparison to the WRF 25km dataset at a longer period, i.e. 20 years instead of 10 years, re-prove the
value from the prediction dataset (see Figure S3). It is acknowledged that the deep-learning prediction is trained with
the reference data (PRISM) first, while dynamical downscaling is a numerical modeling method without the direct
feeding from the reference dataset.

Further, the prediction output also captures the temporal evolution well for both yearly trends and seasonal cycles
(Figure 3). When averaged over the whole western US as the plotted domain in Figure 2, the yearly averages from
coarse-resolution input and the WRF 4km show overestimation (i.e. warmer signal) compared to PRISM, for about
0.6-0.8 K, and 0.3-0.6 K, respectively. The bias from the prediction (i.e. Supres) is reduced with minor warm or cold
differences, ranging from -0.02 to 0.25 K for different years. The seasonal cycle is overall well-captured for all of the
datasets, with some overestimation signal found in the input and the WRF 4km dataset, especially over the summer
seasons (up to 1 K). And the results from prediction closely match the PRISM dataset with bias within 0.3 K. As for
the correlations, it is about ~0.97, 0.94, and 0.96 between Supres and ERA-interim/PRISM/WRF 4km, respectively.
Although Supres correlates better with the driving data given the prediction is directly affected by the inter-annual
variability of the climatology, it is beneficial that prediction can inherit and reflect the temporal variability from the
input but with much finer spatial details.



270

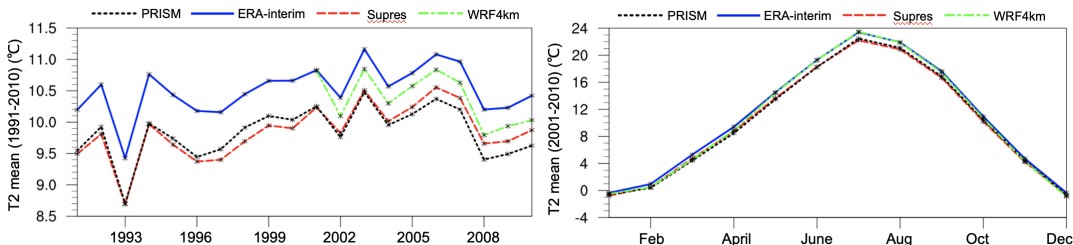

271

**Figure 3: Temporal evolution for T2. Left panel:** Yearly mean over the western US for PRISM, ERA-interim, Supres, and WRF 4km results from year 1991 to 2010 (note that WRF 4km only covers 2001-2010); **Right panel:** Similar as left panel, but for the seasonal cycle (i.e. monthly average) during 2001 to 2010.

In addition to the yearly and monthly average, the fine-scale temperature at daily scale is important in many applications: such as in understanding warming impacts on hydroclimate over complex mountainous regions, and quantification of heatwaves risks. For instance, over the southwest US, the near-surface temperature is generally hotter than other regions with a high risk of heatwaves during hot seasons. To further investigate the performance of the prediction in capturing the daily properties, frequency distributions of the daily T2 values over the southwest region (including California, Nevada, Utah, and Arizona) are exhibited in Figure 4. It is recognizable that the prediction can represent daily distribution well compared to the PRISM, which is comparable to the WRF results with observable improvement from the input.

283

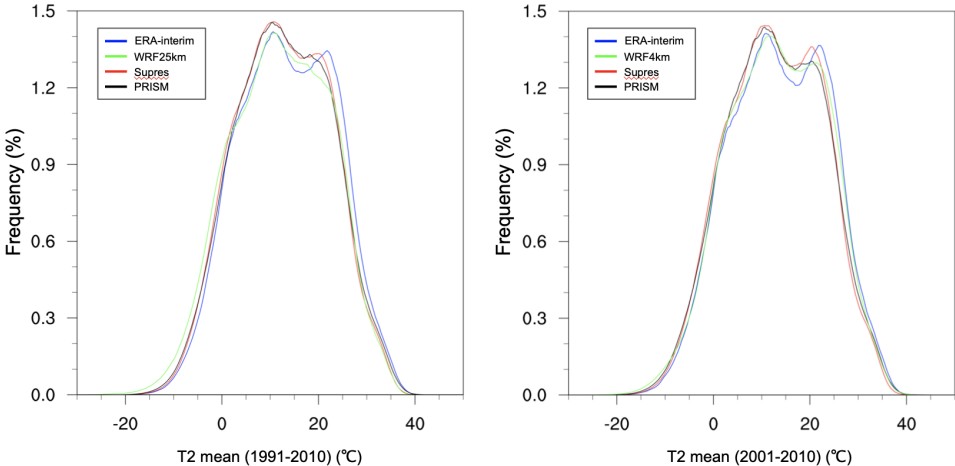

284

**Figure 4: Daily frequency distribution of T2 based on all the grid values during 1991-2010 (left) and 2001-2010 (right)**. The domain covers the southwest states (including California, Nevada, Utah, and Arizona) for results from ERA-interim, Supres, PRISM, and WRF at 25km/4km.

288

289



**3.2 Precipitation**

Unlike temperature, precipitation is non-continuous and is involved with complex regional features, making it much more difficult to downscale for very high-resolution information from a coarse-resolution input. The intrinsic complication of precipitation downscaling requires a well-trained network. As described in the dataset section, additional relevant supporting data to precipitation downscaling include zonal and meridional winds (U and V), relative humidity (RH), and specific humidity (Q) at 850 hPa vertical level from the input are also used. However, precipitation over the western US is still largely controlled by the complex topography and orographic forcings, which also makes the elevation details the key supporting information to reconstruct the spatial details.

Firstly, the yearly mean precipitation is investigated (Figure 5). As observed, the prediction exhibits a similar spatial pattern compared to the PRISM and WRF 4km with significant improvement from the input. The spatial correlation is about ~0.96 between Supres and PRISM, and the input and WRF 4km show a correlation with PRISM for about ~0.89 to 0.93 over the whole domain. The spatial patterns and details are much better represented in the high-resolution output, especially over the west US regions with heavy precipitation. The input overall underestimates the precipitation over most of the regions, especially the heavier precipitated locations for about 2 to 8 mm/day when compared to PRISM. The deep-learning based downscaling output shows significantly reduced biases over the majority of study regions with differences less than 1 mm/day from the reference. Furthermore, the prediction results are comparable to the dynamical downscaling output from WRF 4km, which shows drier bias over the coastal area and wetter biases over the inland regions with biases for about -3 to 3 mm/day. Further comparison to the WRF 25km can be found in Figure S4, which further proves the importance of fine-scale features in retrieving precipitation distributions.

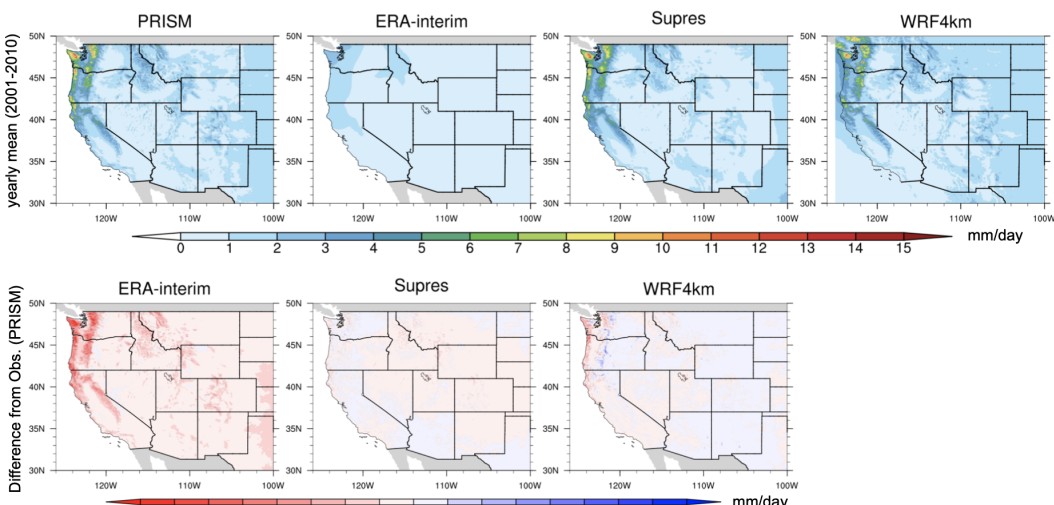

**Figure 5: Yearly mean precipitation over 2001-2010. Upper row:** Precipitation mean from PRISM, ERA-interim, Supres, and WRF 4km; **Bottom row:** Absolute differences from PRISM for the input, prediction, and WRF dynamical downscaling results.



The precipitation features are further examined in terms of yearly trend and seasonal cycle in western US states, where
heavier precipitation locates including WA (Washington), OR (Oregon), and CA (California) (Figure 6). The results
show that prediction from the deep-learning downscaling can represent both the yearly trend and the seasonal cycle
in a reasonable way for all of the three regions, with obvious improvement from the input. Specifically, the yearly
mean values have been underestimated for about 56 to 62% on average over the three regions compared to PRISM in
the ERA-interim, and the biases have reduced to -5% to 2% in the prediction. The WRF 4km results are also close to
the PRISM observations with relative biases for about -2 to -10%. As for monthly mean, the input underestimated the
precipitation on average for about 1.7, 1.24, and 1.0 mm/day for WA, OR, and CA, respectively, with the relative bias
for about ~60% compared to PRISM. The bias has significantly reduced in the fine-scale output with relative biases
for about 8 to 20%. It is further proved that the results from prediction are comparable to WRF 4km, which shows an
average bias for about ~6% to 13% over the different states. (A supplementary comparison to WRF25km across the
longer period (i.e. 1991-2010) can be found in Figure S5.)

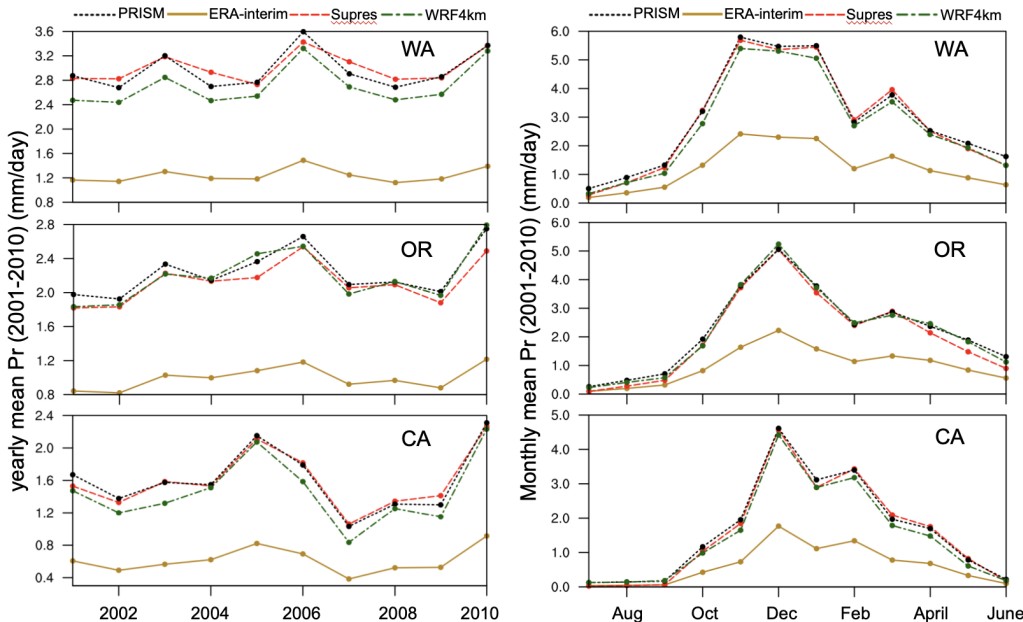

**Figure 6: Time series features for Pr from 2001 to 2010 over the western US states including WA, OR, and CA.**
**Left panel:** Yearly averaged for PRISM, ERA-interim, Supres, and WRF 4km results; **Right panel:** Similar as left
panel, but for the seasonal cycle (i.e. monthly average).

Regions like the western US coast could be significantly impacted by heavy precipitation events. To further investigate
the performance of the prediction in capturing the daily precipitation features, frequency distributions of the daily Pr
over the western US coast (covering WA, OR, and CA) are shown in Figure 7. Here, both the distributions from the
20 years and 10 years period are examined, and the prediction shows an overall good match to the PRISM with notable



improvement from the input. This further proves the added values of the high-resolution dataset in capturing

precipitation extremes (Figure 7). WRF results at the same resolution (i.e. 4 km) show better performance than the

prediction in capturing the extreme daily precipitation values and the distributions. Given the well-reproduced

temperature and precipitation features either for the mean climatology or at the daily scale, the results exhibit that a

robust deep-learning neural network can be used to get high-quality fine-scale climate information alternatively.

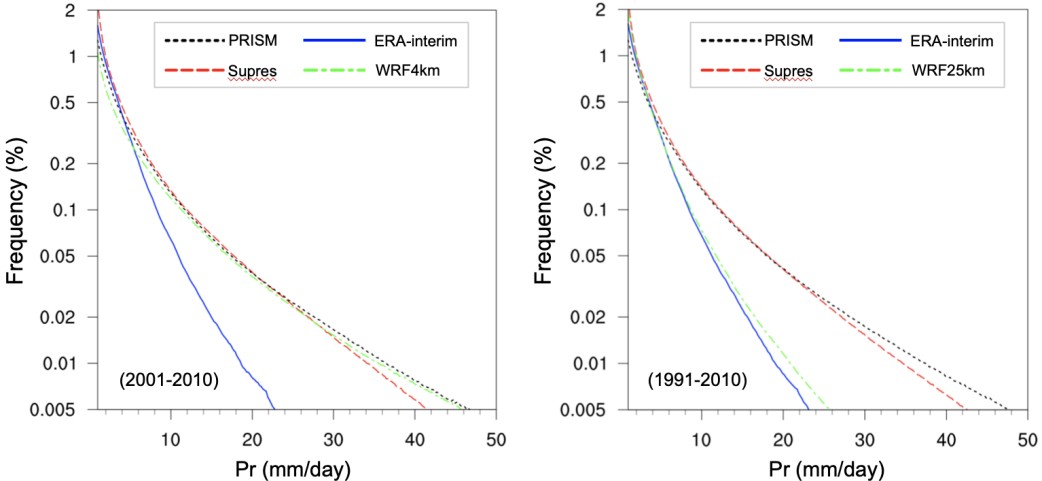

**Figure 7: Daily frequency distribution of Pr based on all the dataset** for 2001-2010 (left) and 1991-2010 (right)

over the grid points from WA, OR, and CA from ERA-interim, Supres, PRISM, and WRF results (note: Y-axis is

logged for better visualization).

**4 Summary and discussions**

In this study, a newly developed deep-learning based framework has been explored for climate downscaling for

temperature and precipitation at a high-resolution of 4 km over the whole western U.S. for 1981 to 2010 at daily scale.

The designed modeling framework, based on a deep-learning super-resolution method, is composed of multiple

convolutional layers, batch normalization, rectification linear unit, and skip connections. In sum, the neural framework

learns optimal parameters from the training data for later-on prediction. Training is based on the first 10 years' daily

input (i.e. 1981-1990) and parameters are optimized using the selected loss function to the "ground-truth" (here,

PRISM gridded observations). The finalized neural network with optimized parameters is then used to perform the

inference over the remaining years' values (i.e. 1991-2010). Given the intuitive attributions of the dataset, the L1 loss

is chosen for temperature and L2 loss is used for precipitation based on the training process and multiple tests.

Results prove that the prediction from deep-learning based downscaling can match the "ground-truth" closely for both

temperature and precipitation. The performance of the deep-learning based modeling framework is also comparable

to traditional regional climate downscaling methods in terms of accuracy in either representing the spatial or temporal



features at a fine grid-scale. The supporting dataset of elevation is key for the neural network to learn orographic
effects, particularly over complex terrains. Precipitation downscaling incorporates additional supporting datasets
including wind and humidity constraints.

This newly developed method and framework largely reduce the time consuming and computation cost for climate
downscaling. PRISM, as a fine-resolution observation dataset, is mainly used for training and validation purposes as
"ground-truth". That is to say, "ground-truth" is only needed in the training stage to optimize the model parameters
and in the inference/testing stage it is not needed. Given a robust deep-learning network developed for downscaling,
a broad application can be further explored including downscaling GCMs' simulations or other types of climate
datasets over an even longer period.

The findings prove that a deep-learning based downscaling method as newly developed here can be a powerful and
effective way to retrieve fine-scale information from a coarse-resolution input. When seeking an efficient and
affordable way for intensive climate downscaling, an optimized convolution neural network framework, as this work
targets, could be an alternative option. The author acknowledges that future work will require a better understanding
of the components in the deep learning methods when applied in climate science.




**Acknowledgements**


The author thanks the helpful comments from editor Fabien Maussion. The author also sincerely thanks Dr. Liu for the enlightening conversations at the study's early stages. The author thanks Dr. Stevenson for helpful comments. The author acknowledges the open-shared dataset used in this study including reanalysis data (ERA-interim), gridded observations (PRISM), and dynamically downscaled data (NA-CORDEX and WRF high-resolution downscaling).

**Author contribution:** This is a single-authored work.

**Code/Data availability:** All post-processed data and codes used in this study can be accessed at the online public portal (https://doi.org/10.5281/zenodo.3996672) (Xingying Huang, 2020) or by contacting the corresponding author (at xingyhuang@gmail.com).

**Competing interests:** The author declares no competing interests.































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
