# Peer review of "Deep-learning based climate downscaling using the super-resolution method: a 2 case study over the western US 3"

_Geoscientific Model Development, 2020_

## Referee Comment (RC1) · Anonymous Referee #1 · 11 Nov 2020

General Comments

This manuscript uses a convolutional neural network (CNN) architecture called Supres (short for Super Resolution) to downscale temperature and precipitation from ERA-Interim resolution (81 km) to PRISM resolution (4 km). The manuscript compares Supres to two WRF simulations, one at 25 km and one at 4 km. Downscaling from an 81 km dataset to a 4 km dataset with CNNs is a very outstanding result; it is furthermore surprising and significant that the neural network only required 9 years of training data for this task. I recommend the paper for publishing after addressing the following comments. My general comments are outlined below:

1) Historical Downscaling: This paper downscales ERA-Interim, which is a historical reanalysis product, using CNNs. In the discussion section, I suggest that the author discuss the scientific significance of downscaling this dataset. What is the benefit of downscaling a purely historical reanalysis product with CNNs? Why not just use PRISM for high-resolution climate data?

a) WRF-based downscaling can be applied to future climate, by forcing WRF with GCMs instead of ERA-Interim. However, CNNs are not as flexible, since a CNN trained on ERA-Interim likely cannot be applied to downscale GCMs simulating future climate. This is because CNNs struggle with extrapolation; a CNN trained on ERA-Interim may not be able to extrapolate to new datasets (e.g. GCMs) that are different from the training dataset (in this case, ERA-Interim). I include more comments about this in #3 below.

b) (Line 371) "a broad application can be further explored including downscaling GCMs' simulations." Because of CNNs' weakness for extrapolating to new datasets, I think it would be more appropriate to use CNNs to downscale WRF-25 km or to postprocess WRF-4km, as there are WRF simulations also available for future climate states.

2) Deep Learning Innovation: In the introduction, the authors present this framework: "Previous studies used mostly basic and early-stage deep learning strategies... the present study aims to incorporate up-to-date deep learning schemes for network de-sign" (Line 62-63, 73-74). Compared to other CNN-based downscaling papers (e.g. Vandal et. al. 2017), it is not clear what the new schemes of Supres are. The Vandal et. al. architecture also leverages ReLU nonlinearities, batch normalization layers, L2 loss, and convolutional layers with a specified stride. Do the new schemes refer to skip connections and the learning rate scheduler? All of the above components are standard components of CNNs.

a) Using the current introduction and framing, I would expect the paper to show how the new aspects of the architecture (for instance, skip connections or a learning rate

scheduler) lead to improvements over existing CNN architectures.

b) Instead, I would recommend changing the introduction to talk in more depth about Empirical Statistical Downscaling (ESD), as CNNs are a form of ESD. I recommend that the author contextualizes CNNs in terms of other ESD algorithms and research, such as bias corrected spatial disaggregation or bias correction and constructed analogs. What are the advantages and disadvantages of CNNs compared to existing methods? I suggest that the authors provide references to ESD algorithms and their discussion.

c) To be clear, I certainly do not think it is necessary for every paper using CNNs to develop a new architecture in order to be published. Rather, I think that this paper should change its framing to emphasize evaluating CNNs in the context of other ESD algorithms and WRF.

3) Stationarity: A common problem with ESD algorithms is that there is uncertainty if the statistical relationship between the low-resolution and high-resolution data will change with climate change. This challenge exists for CNN-based downscaling as well. This problem might be accentuated for CNNs, as their architectures with millions of parameters represent complex, nonlinear functions. Compared to simpler, more interpretable ESD methods (such regression-based methods which use fewer tunable parameters), CNNs may exhibit even more unexpected behavior in a warmer climate and in simulations of a warmer climate. How could the CNN trained in this paper be used in new contexts?

a) Of course, the questions and discussions in the above point do not have to be fully addressed and solved in this paper. However, I think it would be useful to bring them up in the discussion section with respect to future directions of this work. Additionally, I think for this paper, it would be satisfactory to evaluate the neural network on the years 2010-2019 in addition (even though WRF data will be unavailable) to validate whether the performance degrades in recent, warmer years.

4) Bias Correction: In certain sections of the paper, I recommend comparing Supres to

monthly-bias-corrected data from WRF. a) In figure 2 and 5, I recommend bias correcting WRF at each grid point using a 2 year training set and then evaluating WRF and Supres on the remaining 8 years. This would help address the acknowledgment (Lines 255-257) that the neural network is being optimized to match PRISM, while WRF is not being fed the PRISM dataset.

b) In figure 3, 4, and 7, I suggest that the author bias corrects ERA-Interim and WRF4km using a 2 year train set. Alternatively (and equivalently), I suggest that the data in the left plot of Figure 3 be plotted as an anomaly of yearly average temperature.

Specific Comments

1) (Section 2.3.3). Often, when training neural networks on high-resolution climate data, the limiting factor is GPU memory. In this section, I recommend that the author list the GPU memory requirements of Supres, the name of the GPU model (e.g. Volta, Tesla, Titan), and the memory of the GPUs used. Additionally, in the attached code, I see that there is a method for FP16 architectures using the apex Python package's automatic mixed precision capability. For reproducibility, I recommend adding a sentence indicating if the model results shown used mixed precision.

2) (Line 298) I suggest adding a figure to support the statement "elevation details [are] the key supporting information to reconstruct the spatial details." A saliency map indicating the relative importance of the elevation information, compared to other variables, would support this statement. Alternatively, a comparison between two CNNs would also support the statement: one CNN trained with elevation information and one CNN trained without elevation information. If the CNNs' performances are significantly different, then the elevation field is crucial to making the high-resolution maps.

3) (Line 116) "The receptive field is defined as the area where the convolutional filters can influence." I suggest rewording this statement. A convolutional filter influences the whole image, since by definition, the filter is convolved over the whole image. However, in this context, I believe that the intention was to define the receptive field as the 3x3

window of the convolutional kernel.

4) (Figure 4) Why are the x axes labelled "T2 Mean"? If the figure is showing the distribution of daily temperatures, when is temperature being averaged? Should the x axis be changed to "Daily Temperature"?

5) (Table 1) Section 2.2 of the Liu et. al., 2017, paper indicates that the 4 km WRF simulations are forced by ERA-Interim, yet Table 1 indicates that they are forced by NARR. What is the reason for the discrepancy?

6) (Line 252-253) Are these mean absolute error metrics on the test set? Or are they on the entire period (1981-2010)? To be reliable metrics, they should be only calculated from the test set, and if they are only calculated on the test set, I suggest that the author explicitly state this information.

7) (Figure 5) The Supres and WRF figures in the bottom row are very difficult to see. The blue is very light. I suggest changing the scale of the colorbar.

8) (Figure S1) What is the loss referring to here (L1 or L2 loss)? Does this refer to temperature or precipitation? Does the line shown here refer to the training set or the test set?

9) Computational Cost: I think it would be helpful to provide an estimate of the computational cost of running WRF 4 km, for comparison to the deep learning architecture. Additionally, I think it would be helpful to acknowledge that the entirety of the downscaling cannot be achieved in 22.75 milliseconds, as the CNN requires an assimilated grid of climate data from ERA-Interim, which is itself computationally expensive to generate. (Of course, WRF simulations also require ERA-Interim, as they are forced by that dataset, so I agree with the claim that CNNs are net more efficient.)

10) (Line 128-129) "28 layers have been utilized." I recommend that the author summarize how the number 28 is reached. It is stated that the encoder has six convolutional layers and the decoder has six layers (three convolutional and three upsampling). What

are the remaining 16 layers?

11) In the supplemental figures, I suggest adding a spatial plot showing the average RMSE at each grid cell across all days in the test set.

12) (Figures 4 and 7) Might I suggest adding figures that are analogous to Figure 5A of Vandal et. al.? Figure 5A of Vandal et. al. directly evaluates the CNN for extreme values of the climate variable. I think this would more directly determine whether the CNN performs well during extreme days, compared to the existing plots showing distributions.

Technical Corrections

(Line 51) Change "generally be constrained" to "generally is constrained" (Line 107) It appears as if the subscripts for the first two summations are missing (Line 121) Typo: it should be "lower dimensional features" rather than "smaller dimension features" (Line 211) "rest" should be "remaining" (Line 222) I think the sentence should be rephrased to "The neural network has approximately 7,500,000 trainable parameters" (Line 228) "depending on what types of GPUs to use" should be changed to "depending on the GPU" (Figures 2, 5) I recommend that the top row be ordered in the order "ERA-Interim, Suppress, WRF 4 km, PRISM." That way, each column refers to the same dataset. (Line 298) I suggest changing "spatial details" to "high-resolution spatial maps" (Line 75) I suggest changing "high-performed GPUs" to "high-performance GPUs" (Line 137-138) I suggest rewording the existing sentence to "In general, a successful deep learning framework must approximate the complex relationship between low-resolution and high-resolution climate variables (such as temperature and precipitation)." The existing wording makes it appear as if the architecture is finding the relationship between temperature and precipitation. (Line 118) What does it mean to "accumulate the receptive field"? If the receptive field is defined as the window of the convolutional kernel (i.e. 3x3), how would adding more convolutional layers or changing the stride affect it?

[Figure]

2020.

---

## Referee Comment (RC2) · Anonymous Referee #2 · 28 Dec 2020

**Overall comments**

The author proposes a deep learning (DL)-based approach for super resolution (SR) in this article for climate downscaling. While the topic is an important one and the paper is reasonably well-written, the analysis is incomplete and insufficient for evaluating the DL model performance satisfactorily. The emphasis on the novelty of the DL approach is disproportionate, especially considering that the model architecture and training procedure are fairly standard. Furthermore, several recent DL-based SR approaches in the climate science literature are missing, which suggests that the author is not up-to-speed on the advancement of this field in the last couple of years. This is reflected in

the lack of breadth and depth of their analyses. As such, this paper is not ready for publication in GMD and will require more in-depth analysis and more critical evaluation of the results. Specific comments and suggestions are below.

Abstract

L14: "the" DL-based SR method? There are many, which one is being referred to here. L16: Only spatial SR is being done should be mentioned here in the abstract and in the body of the manuscript L17: It does not appear that there is anything new in this framework in terms of the DL architecture. Could the authors clearly state what is new and why they consider it "new"? L23-25: Some details can be given on why the DL model performance is competitive L26: It would help to be more specific about what types of coarse data this model can be used for, presumably it will not work well for all types of coarse climate data L28: In order to be applicable broadly, the DL model needs to be tested for generalization. Evidence of such analyses could be presented briefly in the abstract.

Body of the manuscript

L56: Some key references, including more advanced DL-based SR frameworks for climate downscaling, are missing: 1. Groenke et al. (arXiv:2008.04679ă[cs.CV]) ClimAlign: Unsupervised statistical downscaling of climate variables via normalizing flows 2. Bano-Medina et al, https://doi.org/10.5194/gmd-2019-278 3. https://link.springer.com/article/10.1007/s00704-020-03098-3 4. https://rmets.onlinelibrary.wiley.com/doi/10.1002/joc.6769 5. Stengel et al., https://www.pnas.org/content/117/29/16805.short 6. Sha et al., https://journals.ametsoc.org/view/journals/apme/59/12/jamc-d-20-0057.1.xml 7. Sha et al., https://journals.ametsoc.org/view/journals/apme/aop/JAMC-D-20-0058.1/JAMC-D-20-0058.1.xml 8. Chen et al., https://arxiv.org/abs/2012.09700 9. Liu et al., https://doi.org/10.1145/3394486.3403366

L62-64: This statement is not true; see the refs above

L73-75: These and other advanced techniques have been used in other studies not referenced in this article. The so-called "up-to-date" and "cutting-edge" DL schemes used in this article are actually quite commonplace. In fact, current DL-based SR strategies in computer vision are more advanced than proposed in this article. c.f. Z. Wang, J. Chen and S. C. H. Hoi, "Deep Learning for Image Super-resolution: A Survey," inÂăIEEE Transactions on Pattern Analysis and Machine Intelligence, doi: 10.1109/TPAMI.2020.2982166.

Sub-sections 2.1 and 2.2 can be compressed into a few key points. These are fairly standard DL methods, which is now quite well-adopted in the climate science community, hence getting into great depth is not warranted for a research article. Citing the key papers from the DL community and describing the chosen DL architecture briefly is sufficient. There are no novelties in these sections.

Table 1 is a helpful summary of the datasets used.

L207-208: Regridding using bilinear interpolation introduces a host of other challenges, in contrast to those from sophisticated super-resolution techniques. Hence the issues introduced by regridding need to be discussed.

L216: The training, validation and testing loss curves need to be shown to convince the reader that the model has converged and overfitting is not an issue — not just the training loss curve. Further, the X axis of the loss curve is typically the number of epochs, not the number of iterations.

Section 3.1: The goal of SR is to capture fine scale details at daily or sub-daily timescales, hence showing the average over 10 years is not useful. As such, Figure 2 and Figure S3 are weak comparisons and do not show the power or limitations of the proposed DL SR model.

Figure 3 (left) appears to show that a simple bias correction of the ERA-I would shift the temporal evolution curve to be closer to the PRISM data. While Figure 3 (right) shows

[Figure]

that the seasonal cycle is almost identical for all models. What is the added value of a complex DL SR model here? Once again, it appears that testing annual averages is not very useful for an SR method, which aims to provide high spatial and temporal resolution. More useful tests include examining the spatial and temporal spectra; spatial and temporal coherent structures; extreme events (tails of the probability distributions of the physical variables downscaled); scatter plots or 2D PDFs to show false positives and false negatives.

Figure 4 is helpful to see differences at the daily timescale but should also be plotted on a log scale to be able to see the differences in the tails (extremes). Further, in depth examination of the extremes is needed to highlight the differences and plausible reasons for the differences.

Similar comments from Figure 3 for Figure 5 and 6 and S4 and S5. 10-year averages tend to smooth out errors in the SR process. Once again, it appears that testing annual averages is not very useful for an SR method, which aims to provide high spatial and temporal resolution. More useful tests include examining the spatial and temporal spectra; spatial and temporal coherent structures; extreme events (tails of the probability distributions of the physical variables downscaled); scatter plots or 2D PDFs to show false positives and false negatives.

Conclusions

L350: the novelties of the DL model shown here do not warrant this statement. L357-358: it is unclear what intuition was used in choosing the L1 vs L2 losses. L360: this is too strong a statement and I don't think the results prove that, certainly not at fine spatial and temporal scales. L363: The role of elevation as an additional feature could be highlighted by training with and without this field to show the differences. This comparison would help make the usefulness of elevation more apparent. L365: Similarly for precipitation, ablation studies with and without the additional physical variables is needed to show their benefit. L374: Generalization studies are needed to make such

strong statements. As such, the results shown do not warrant this statement.